# Carboxylic acid-modified metal oxide catalyst for selectivity-tunable aerobic ammoxidation

Xiuquan Jia[1], Jiping Ma[1], Fei Xia[1,2], Yongming Xu[1,2], Jin Gao[1] & Jie Xu ![ORCID] [1]

Controlling the reaction selectivity of a heterobifunctional molecule is a fundamental challenge in many catalytic processes. Recent efforts to design chemoselective catalysts have focused on modifying the surface of metal nanoparticle materials having tunable properties. However, precise control over the surface properties of base-metal oxide catalysts remains a challenge. Here, we show that green modification of the surface with carboxylates can be used to tune the ammoxidation selectivity toward the desired products during the reaction of hydroxyaldehyde on manganese oxide catalysts. These modifications improve the selectivity for hydroxynitrile from 0 to 92% under identical reaction conditions. The product distribution of dinitrile and hydroxynitrile can be continuously tuned by adjusting the amount of carboxylate modifier. This property was attributed to the selective decrease in the hydroxyl adsorption affinity of the manganese oxides by the adsorbed carboxylate groups. The selectivity enhancement is not affected by the tail structure of the carboxylic acid.

---

[1] State Key Laboratory of Catalysis, Dalian National Laboratory for Clean Energy, Dalian Institute of Chemical Physics, Chinese Academy of Sciences, Dalian 116023, China. [2] University of Chinese Academy of Sciences, Beijing 100049, China. Correspondence and requests for materials should be addressed to J.M. (email: majiping@dicp.ac.cn) or to J.X. (email: xujie@dicp.ac.cn)

Heterogeneous catalysis is an essential technology for developing benign processes in the chemical industry. In addition to high activity, heterogeneous catalysis should offer excellent selectivity for the desired product, thereby minimizing the costs associated with product purification and waste disposal. In particular, an enhanced chemoselectivity is of primary importance in the processing of polyfunctional molecules[1–6]. However, irregular catalyst surfaces commonly contain diverse active sites, resulting in a variety of undesirable reaction pathways. In recent years, surface modification has been clearly recognized as an ideal technique for controlling the selectivity, enabling an enhanced reactant orientation or fine-tuning of the active sites on the catalyst surface[7–12]. Owing to their flexibility and versatility, organic modifiers represent an important set of compounds for the surface modification of noble metal nanoparticles[11]. Medlin's group has successfully improved the selectivity of the hydrogenation of 1-epoxy-3-butene to 1-epoxybutane by modifying the conventional Pd/Al$_2$O$_3$ catalyst with alkanethiol[13]. Later, a series of pioneering studies were reported on self-assembled thiol and amine modifiers that enhanced the chemoselective reaction of bifunctional compounds, for example, the chemoselective nitrostyrene reduction[14], hydrogenation of cinnamaldehyde to cinnamyl alcohol[15,16], hydrogenation of the aldehyde moiety of furfural to produce furfuryl alcohol and methylfuran[17–19], and, more recently, hydrogenation of acetophenone to phenylethanol[20]. Although a variety of organic modification strategies have successfully improved the chemoselective reactivity, the application of this strategy has been limited to the confined catalyst field of metallic nanoparticles, especially noble metal nanoparticles[21]. Only a few reports are available on the effective modification of base-metal oxide catalysts with an organic modifier[22,23], which should be another potentially fruitful route to practical chemoselective redox chemistry because metal oxides are useful catalysts in many fields of chemistry owing to their redox properties with high stability and durability.

In heterogeneously catalyzed reactions, the diffusion, chemisorption, and surface reaction of the reactant are three elementary processes that affect the reaction selectivity[24,25]. The reported organic modification strategies have enhanced the selectivity of metal nanoparticle catalysts by favorably influencing the diffusion or surface reaction step[11,16,17,26–28]. Despite the numerous reports investigating the chemisorption behaviors of organic molecules, the effect of organic modification on the reactant chemisorption affinity of a catalyst is not clear. This relationship remains important for the further development of this catalyst modification technique. In addition, quantitatively tuning the adsorbate affinity of the catalyst surface would strongly aid in determining the relationship between the reactant adsorption behavior and the corresponding reaction pathways[11]. Consequently, from the viewpoint of practical and mechanistic chemistry, exploring the regulation of the reactant adsorption affinity of metal oxides by pre-adsorption of organic modifiers to control

the reaction selectivity is worthwhile. Recently, our group focused on controlling the oxidation selectivity of manganese oxide (MnO$_x$) catalysts[29–31].

In this article, we directly modify the adsorption affinity of MnO$_x$ catalysts with environmentally friendly carboxylates. The carboxylic acid modifiers control the alcohol adsorption affinity of the catalyst, resulting in tunable alcohol oxidation reactivity. MnO$_x$ modified with carboxylic acids exhibits chemoselectivity in hydroxynitrile synthesis via aerobic ammoxidation of hydroxyaldehyde.

## Results

**Catalyst synthesis and characterization.** The dissociative adsorption of an alcohol reactant on a catalyst is generally believed to be an important step in the subsequent aerobic oxidation reaction. According to a previous report by Tamura and co-workers[32], good linearity exists between the oxidation reactivity of an alcohol and the amount of on-top and bridged sites available for the dissociative adsorption of hydroxyl species on a CeO$_2$ catalyst. The O–H bond dissociation of hydroxyl and carboxyl groups is normally activated by similar sites on metal oxides[33–35]. Thus, the chemisorption behavior of alcohol can reasonably be altered via carboxylate modification of the catalyst surface, thereby further tuning the oxidation reactivity and product selectivity of the catalyst (Fig. 1).

Carboxylic acid-modified MnO$_x$ were prepared by deposition from a dilute solution in aprotic solvent (CH$_3$CN) according to modification of a previously described method[36,37]. When deposited on a metal oxide surface from a dilute solution, these carboxylic acids spontaneously arrange to form a self-assembled interspersion[38]. To examine the effect of carboxylic acid surface modification on the chemisorption of alcohol on MnO$_x$, in situ Fourier transform infrared (FT-IR) spectroscopic characterization of CH$_3$OH adsorption (Fig. 2c) was performed. Methanol dissociates into a methoxy species on unmodified MnO$_x$, and the bands of the methoxy species observed at 1090 and 1028 cm$^{-1}$ are assigned to the $\nu$(C–O) bands of the on-top and bridged sites, respectively (Fig. 2a)[32,33]. Under identical conditions, the C–O vibrations arising from the methoxy species were barely observed for the acetic acid (HAc)-modified MnO$_x$. The effect of carboxylic acid surface modification on the adsorption of benzyl alcohol on MnO$_x$ was also investigated with FT-IR spectroscopy by adding benzyl alcohol to MnO$_x$ followed by evaporation of the unadsorbed species (Supplementary Figs. 1 and 2). The influence of the surface modification on the adsorption of benzyl alcohol was the same as that of methanol. These results demonstrate that the sites for the dissociative adsorption of alcohols can be blocked by pre-adsorption of a carboxylate modifier.

Next, the mode of adsorption site blockage upon surface modification was studied. The self-assembly of different amounts of HAc on the surfaces of manganese oxides was characterized by FT-IR spectroscopy (Fig. 2d). Spectra were collected after

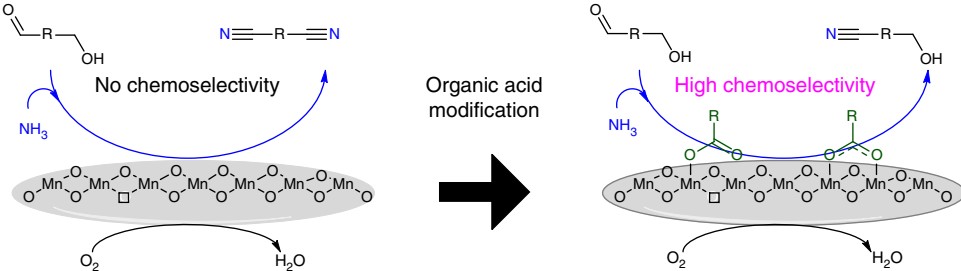

**Fig. 1** Design strategy for controlling the reaction selectivity over MnO$_x$ catalysts. Schematic illustration of the strategy for achieving the selectivity-tunable aerobic ammoxidation of hydroxyaldehyde by an organic carboxylic acid-modified MnO$_x$ catalyst

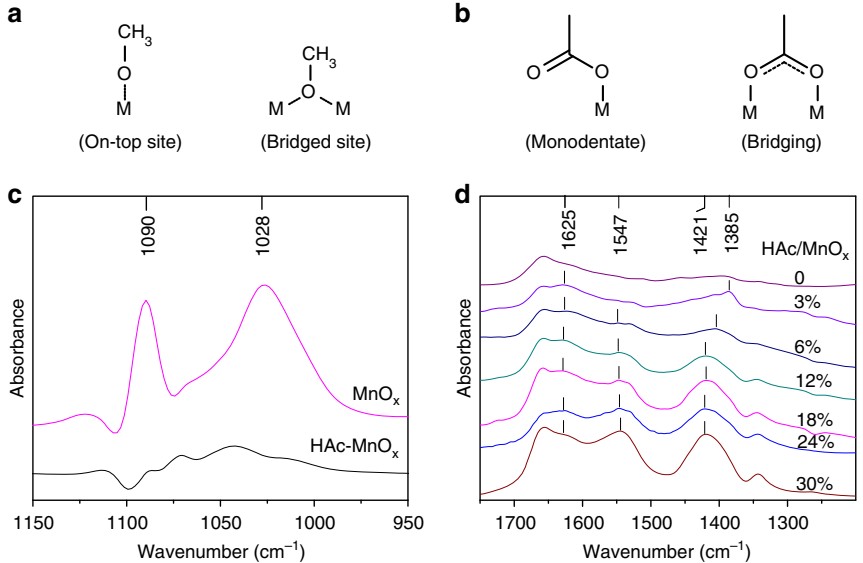

**Fig. 2** FT-IR spectra of unmodified and HAc-modified $MnO_x$. **a** Model of the sites for $CH_3OH$ adsorption on metal oxides. **b** Coordination modes of the carboxylate adspecies. **c** FT-IR spectra of $CH_3OH$ adsorption to unmodified and HAc-modified $MnO_x$. **d** FT-IR spectra of $MnO_x$ according the corresponding spectrum. The Mn content was analyzed by inductively coupled plasma atomic-emission spectroscopy

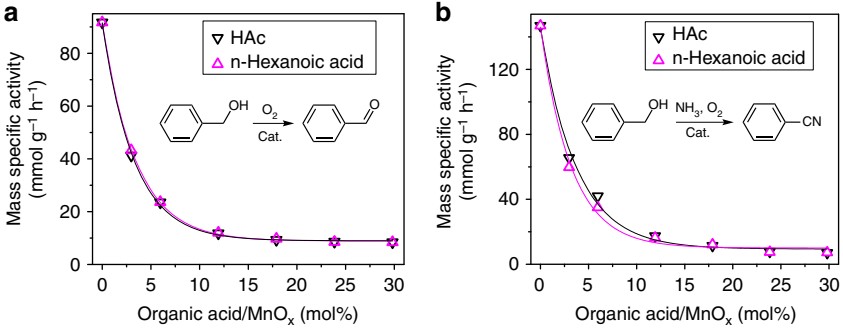

**Fig. 3** Oxidative activities of unmodified and HAc-modified $MnO_x$. **a** Mass-specific activity (mmol $g_{cat}^{-1}$ $h^{-1}$) of $MnO_x$ in benzyl alcohol oxidation as a function of the organic acid/$MnO_x$ ratio. **b** Mass-specific activity (mmol $g_{cat}^{-1}$ $h^{-1}$) of $MnO_x$ in benzyl alcohol ammoxidation as a function of the organic acid/$MnO_x$ ratio. The Mn content was analyzed by inductively coupled plasma atomic-emission spectroscopy. The mass-specific activity was measured below 30% conversion

removing the solvent by heating the sample at 80 °C. For both the pure and modified $MnO_x$ samples, broad peaks were observed at ~1655 cm$^{-1}$, which cannot be attributed to HAc adsorption. After treatment with a dilute HAc solution (HAc/$MnO_x$ = 3 mol %), two bands were observed at 1385 and 1625 cm$^{-1}$ that arise from the symmetric and asymmetric stretching of the carboxylate groups ($v_s$ and $v_{as}$, respectively). The difference ($\Delta_{as-s}$) between these two bands ($v_{as}$ and $v_s$) is 240 cm$^{-1}$, suggesting a monodentate mode (Fig. 2b, Supplementary Fig. 3) of (–COO)/Mn binding to the on-top sites[39–41]. With an increase in the amount of HAc from 6 to 30 mol%, a blueshift in the symmetric stretching vibration band from 1404 to 1421 cm$^{-1}$ was observed. Simultaneously, another asymmetric stretching vibration band ranging from 1549 to 1544 cm$^{-1}$ was observed, and this band had a $\Delta_{as-s}$ <150 cm$^{-1}$, indicative of a bridging mode (Fig. 2b and Supplementary Fig. 3) of (–COO)/Mn coordination, which should occur at the bridged sites on the $MnO_x$ surface[39–41]. Thus, we deduce that, during the self-assembly process, the adsorption of carboxylate groups on $MnO_x$ preferentially occurred through monodentate binding, blocking the on-top sites, and then through bidentate coordination to the bridged sites, blocking bridging mode adsorption.

**Fig. 4** Aerobic ammoxidation of 4-hydroxymethylbenzaldehyde. The ammoxidation of **1** can occur selectively on the aldehyde group to give hydroxynitrile product of **2**, or can occur nonselectively to give dinitrile product of **3**

If the amount of alcohol that dissociatively adsorbs on a metal oxide surface can be controlled, the aerobic oxidation activity of the catalyst can be intentionally tuned. We thus investigated the relationship between the carboxylate modification of $MnO_x$ and the catalytic performance of the resulting catalysts in benzyl alcohol oxidation/ammoxidation (Fig. 3a, b). Unmodified $MnO_x$ showed superior activity based on the benzyl alcohol conversion (92 mmol $g_{cat}^{-1}$ $h^{-1}$ for aerobic oxidation, 147 mmol $g_{cat}^{-1}$ $h^{-1}$ for aerobic ammoxidation). Notably, the HAc-modified catalysts exhibited a dramatic decrease in activity in both alcohol oxidation and ammoxidation. In the initial stage, the decrease in activity remained even with the increase in the amount of carboxyl groups without the observation of a delay period. This result

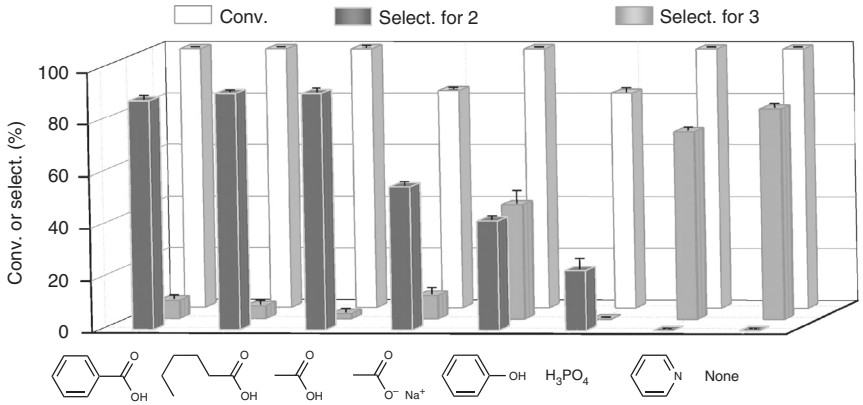

**Fig. 5** Performance of MnO$_x$ with different modifiers in the ammoxidation of 4-hydroxymethylbenzaldehyde. Reaction conditions: 0.5 mmol 4-hydroxymethylbenzaldehyde, 0.17 mmol MnO$_x$, additive/MnO$_x$ = 120 mol%, 5 mL CH$_3$CN, 0.3 MPa NH$_3$, 0.3 MPa O$_2$, at 80 °C for 3 h

**Fig. 6** Chemoselective ammoxidation of 5-hydroxymethylfurfural. The HAc-modified MnO$_x$ catalyst is effective for the selective ammoxidation of 5-hydroxymethylfurfural to hydroxynitrile

**Table 1 Performance of manganese oxides with different amount of HAc modifiers in ammoxidation of 4-hydroxymethylbenzaldehyde**

| Entry | HAc/MnO$_x$ (mol%) | Conv. (%) | Select. (%) | |
|---|---|---|---|---|
| | | | 2 | 3 |
| 1[a] | 0 | >99 | 0 | 81 ± 1 |
| 2 | 6 | >99 | 12 ± 3 | 66 ± 1 |
| 3 | 12 | >99 | 45 ± 1 | 41 ± 3 |
| 4 | 30 | >99 | 71 ± 2 | 25 ± 2 |
| 5 | 120 | >99 | 91 ± 1 | 2 ± 1 |
| 6 | 300 | >99 | 92 ± 1 | 1 ± 0.4 |

Reaction conditions: 0.5 mmol 4-hydroxymethylbenzaldehyde, 0.17 mmol MnO$_x$, 5 mL CH$_3$CN, 0.3 MPa NH$_3$, 0.3 MPa O$_2$ at 80 °C for 3 h
[a]18 ± 1% 4-Cyanobenzamide was detected

indicates that modification of the surface with carboxylates directly suppressed the alcohol oxidation/ammoxidation reactivity. As the packing of HAc molecules became increasingly dense, the rate of oxidation decreased by more than ten times (from 92 to 8 mmol g$_{cat}^{-1}$ h$^{-1}$), and the rate of ammoxidation decreased by over 20 times (from 147 to 7 mmol g$_{cat}^{-1}$ h$^{-1}$). n-Hexanoic acid showed an effect similar to that of HAc, thus excluding the possible steric hindrance effect of the tail moiety of the carboxylate modifier.

Moreover, the observed nonlinear correlation indicates that the activities of the modified MnO$_x$ were strongly related to the adsorption mode of the modifiers. When the adsorption mode of HAc was monodentate, the mass-specific activity rapidly decreased as the HAc content increased (from 92 to 41 mmol g$_{cat}^{-1}$ h$^{-1}$ for aerobic oxidation, from 147 to 65 mmol g$_{cat}^{-1}$ h$^{-1}$ for aerobic ammoxidation). In comparison, the adsorption of HAc through the bridging mode resulted in only a slight decrease in activity. The above results reveal that there is a considerable difference in activity between the on-top and bridged sites on the MnO$_x$ surface. Most importantly, we demonstrated that the reactivity in alcohol oxidation/ammoxidation can be controlled through carboxylate modification of the MnO$_x$ surface.

**Catalytic chemoselective reactions**. The direct synthesis of nitriles or amides from alcohols or aldehydes by aerobic ammoxidation using MnO$_x$ catalysts has been recently achieved[31,42–44]. Hydroxyaldehydes constitute a class of important chemicals in nature[45–48], and their oxidative conversion while preserving the active hydroxyl group is challenging. To date, a general procedure for the selective ammoxidation of unprotected hydroxyaldehyde to hydroxynitrile compounds, which are compounds plants secrete to protect against herbivore attacks[49–51], has not yet been reported.

The aerobic ammoxidation of 4-hydroxymethyl-benzaldehyde was selected as a model reaction (Fig. 4). In the next experiments, different modifiers were used, and the effects of the modifiers on the catalytic performance of the MnO$_x$ were monitored (Fig. 5). Using only MnO$_x$ without a modifier produced the products of

complete oxidation/ammoxidation of dinitrile and cyanoamide, showing no chemoselectivity. By adjusting the reaction conditions, including the reaction time, reaction temperature, and ratio of the substrate to the catalyst, approximately 30% of the chemoselective hydroxynitrile ammoxidation product was obtained as the highest yield (Fig. 7a and Supplementary Figs. 4 and 5). Modification with pyridine minimally affected the selectivity relative to that of clean MnO$_x$. The modification of MnO$_x$ with sodium acetate or phosphoric acid significantly lowered the activity and the carbon balance due to undetected side reactions. Modification with phenol gave moderate selectivity for hydroxynitrile along with a considerable amount of the dinitrile product. In contrast, benzoic acid, n-hexanoic acid, and acetic acid provided much higher chemoselectivities than the other modifiers. The MnO$_x$ modified by carboxylic acids with different carbon chains produced hydroxynitrile with a similar selectivity of approximately 90%. These results suggest that the tuning of the selectivity is not affected by the tail moiety of the carboxylic acid modifier. Thus, the effect of the modifier on the selectivity stems exclusively from the adsorption of the carboxylate, which selectively lowers the hydroxyl group affinity of the catalyst. Aldehyde ammoxidation was not obviously affected by the presence of a modifier, which may be due to the large difference in the chemisorption behaviors of aldehyde and aldimine intermediates on metal oxides compared to that of alcohol.

This approach is also effective for the construction of hydroxynitrile from biomass-derived hydroxyaldehyde. 5-Hydroxymethylfurfural was selectively transformed to 5-hydroxymethylfuronitrile by the HAc-modified MnO$_x$ catalyst (Fig. 6).

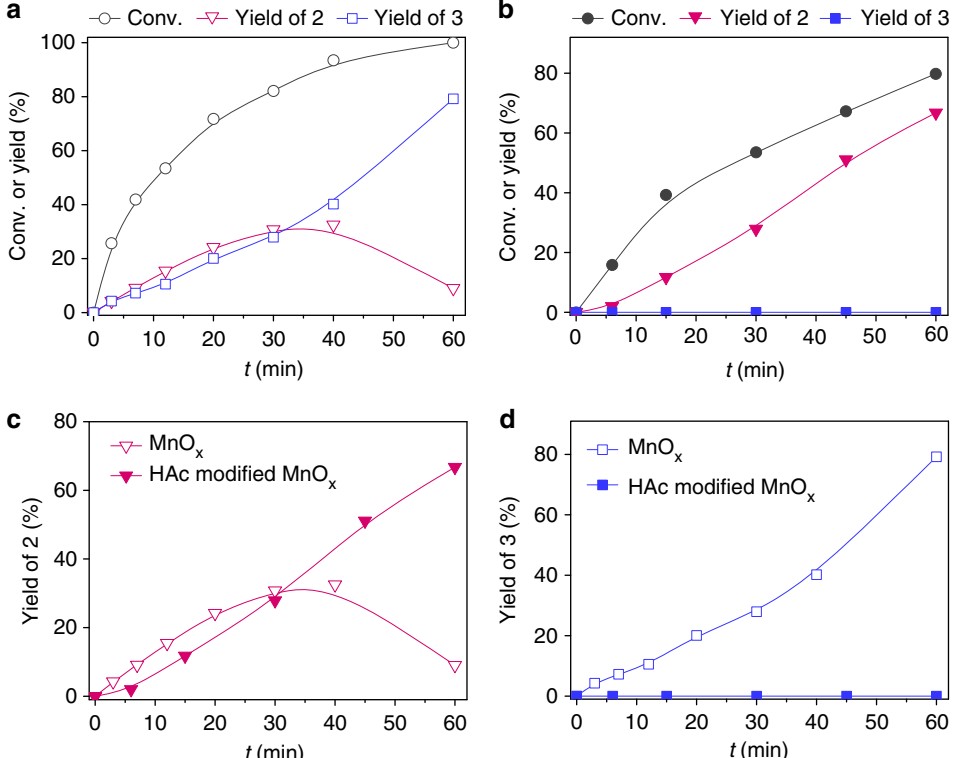

**Fig. 7** Time course of the aerobic ammoxidation of 4-hydroxymethylbenzaldehyde. **a** $MnO_x$ as catalyst. **b** HAc-modified $MnO_x$ as catalyst. **c** Time course of **2** formation over $MnO_x$ and HAc-modified $MnO_x$, respectively. **d** Time course of **3** formation over $MnO_x$ and HAc-modified $MnO_x$, respectively. Reaction conditions: 0.5 mmol 4-hydroxymethylbenzaldehyde, 0.17 mmol $MnO_x$, HAc/$MnO_x$ = 120 mol% for HAc-modified $MnO_x$, 5 mL $CH_3CN$, 0.3 MPa $NH_3$, 0.3 MPa $O_2$ at 80 °C

To obtain a better understanding of the role of the carboxylic acid in the ammoxidation reaction catalyzed by $MnO_x$, we analyzed the correlation between the amount of carboxylic acid and the catalytic performance of $MnO_x$ in the ammoxidation of 4-hydroxymethylbenzaldehyde (Table 1). Unmodified $MnO_x$ showed superior activity in the conversion of the hydroxymethyl group as well as the aldehyde group, producing the products of complete ammoxidation of dinitrile and cyanoamide. The product distribution changed once acetic acid was introduced. When the carboxylate concentration increased, a steady increase in the selectivity for hydroxynitrile was observed. In addition, the same trend was observed at 90, 70, and 50% conversion (Supplementary Fig. 6). As the carboxylate groups on the surface became increasingly dense, the activity of the catalyst in alcohol oxidation was suppressed, providing enhanced selectivity for hydroxynitrile.

While the ability to achieve high selectivity at full conversion is significant, the reaction kinetics should also be investigated to provide insight into the ammoxidation activity of the catalyst. As shown in Fig. 7a, the initial stage of reaction gave comparable amount of hydroxynitirle and dinitrile over unmodified $MnO_x$. By contrast, dinitrile was not detected over HAc-modified $MnO_x$ (Fig. 7b, d), indicating an extremely low rate of dinitrile formation. However, the rate of hydroxynitirle production was slightly decreased after modification (Fig. 7c). Thus, the enhanced chemoselectivity did not result from an increase in the rate of the desired reaction by the modified catalyst but rather from the inhibition of undesired reactions.

Further studies on the stability of hydroxynitrile under the oxidative conditions were performed. An initial gradual increase in the selectivity for hydroxynitrile was observed owing to the formation and oxidation of the aldimine intermediate. After

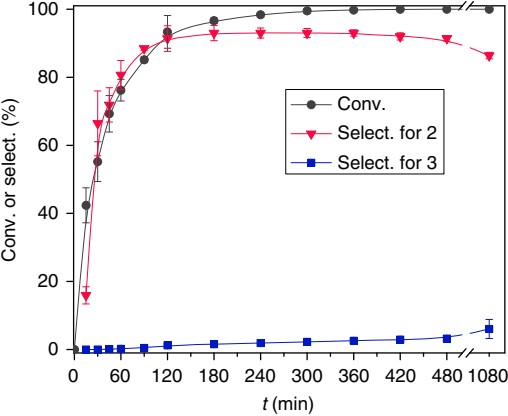

**Fig. 8** Time course of the catalytic conversion of 4-hydroxymethylbenzaldehyde over HAc-modified $MnO_x$. Reaction conditions: 2.5 mmol 4-hydroxymethylbenzaldehyde, 0.85 mmol $MnO_x$, HAc/$MnO_x$ = 300 mol%, 25 mL $CH_3CN$, 0.3 MPa $NH_3$, 0.3 MPa $O_2$ at 80 °C

complete conversion of the hydroxyaldehyde, the selectivity of the reaction for hydroxynitrile remained above 90%, even after extending the reaction time to 8 h (Fig. 8). Furthermore, the concentration of dinitrile remained very low. When the reaction time was extended to 18 h, the selectivity for hydroxynitrile decreased slightly. The ability to prevent alcohol ammoxidation over a long period suggests that the regulated chemisorption properties of the carboxylate-modified catalyst surface are not easily destroyed under the reaction conditions. Moreover, the catalyst can be recycled without loss of selectivity through

regeneration (Supplementary Fig. 7). The development of an improved recycling procedure is in progress.

## Discussion

In summary, we have shown that carboxylate-modified $MnO_x$ can be used as an efficient catalyst for the selective ammoxidation of hydroxyaldehyde under mild conditions. Detailed studies revealed that carboxylic acid modification is capable of regulating the alcohol chemisorption affinity of $MnO_x$, resulting in a tunable distribution of the dinitrile and hydroxynitrile products. The presence of adsorbed carboxylate on the surface of the $MnO_x$ can selectively block hydroxyl adsorption, which is essential for achieving a high chemoselectivity for hydroxynitrile. Carboxylic acids with different side chains exhibited comparable selectivities. Thus, the self-assembled modification of metal oxides can potentially serve as a complement to recent efforts aimed at controlling the selectivity of oxidation and ammoxidation reactions. Studies on the intrinsic mechanism leading to the controlled selectivity of the organic carboxylic acid-modified $MnO_x$ toward hydroxynitrile synthesis are underway.

## Methods

**Preparation of $MnO_x$.** A 100 mL aqueous solution containing 40 mmol of $KMnO_4$ was added into another 250 mL solution of $EtOH$-$H_2O$ (5:1) containing 40 mmol of $MnAc_2$. After the addition was complete, the pH was adjusted to 8 with aq. $NH_3$, and then the mixture was stirred at room temperature for 12 h. The resulting solid was collected by filtration, washed repeatedly with distilled water, and dried for 48 h in air at 80 °C.

**Preparation of organic acid-modified $MnO_x$.** The organic acid-modified catalysts were prepared by immersing the catalyst in an acetonitrile solution of the modifier at 30 °C. After stirring for 3 days, the mixture containing the modifier solution and the catalyst was used in reactions without further separation.

**Catalyst characterization.** X-ray powder diffraction (XRD) patterns were obtained using a Rigaku D/Max 2500/PC powder diffractometer with Cu Kα radiation ($\lambda = 0.15418$ nm) operated at 40 kV and 200 mA with a scanning rate of $5°$ min$^{-1}$ (Supplementary Fig. 8). In situ FT-IR spectra of the methanol adspecies on the catalysts were recorded with a TENSOR 27 spectrometer equipped with an in situ infrared (IR) cell connected to a conventional gas flow system. The samples (20–30 mg) were pressed into self-supporting wafers (20 mm in diameter) and mounted in the IR cell. The adsorption of methanol was carried out with the following method: The sample was pretreated at 150 °C under vacuum (<10$^{-1}$ Pa) for 30 min. After cooling to 30 °C, methanol was fed into the in situ IR cell under vacuum (<10$^{-1}$ Pa). Then, the IR disk was purged with $N_2$, and IR measurements were carried out until the spectrum was stable. The HAc adspecies on the catalysts were also recorded by FT-IR with the TENSOR 27 spectrometer. Before the experiment, the $MnO_x$ sample was immersed in an acetonitrile solution of HAc at 30 °C and stirred for 3 days. Then, the suspension was added dropwise onto the surface of a self-supporting KBr wafer (100 mg, 20 mm in diameter) and dried at 80 °C for approximately 30 min to evaporate the liquid. IR measurements were carried out to obtain the signal of the adsorbed substrates. The benzyl alcohol adspecies on the catalysts were recorded in a similar way except that drying of the added suspension was performed for 12 h to evaporate the unadsorbed benzyl alcohol.

**General experimental procedures.** The catalytic reactions were performed in a 20 mL stainless-steel autoclave equipped with a magnetic stirrer, a pressure gauge, and an automatic temperature control apparatus. The reactor was connected to an oxygen cylinder for adjusting the reaction pressure. In a typical experiment, 4-hydroxymethylbenzaldehyde (62.4 mg, 0.5 mmol) and the prepared suspension of organic acid-modified $MnO_x$ were loaded into the reactor. After sealing the reactor and charging it with $NH_3$ (0.3 MPa) and $O_2$ (0.3 MPa), the autoclave was heated to the desired temperature (80 °C). After reaction, the autoclave was cooled, and the solution was separated by centrifugation and analyzed by gas chromatography (GC) using the internal standard method. (Supplementary Fig. 9) The error bars (standard deviation) in the selectivities were calculated from repeat measurements.

The products were identified by an Agilent 6890 N GC/5973MS and by comparison of the retention times to the corresponding standards in the GC trace (Supplementary Figs. 10–12). GC measurements were conducted on an Agilent 7890 A GC with an autosampler and a flame ionization detector. A DB-17 capillary column (30 m × 320 μm × 0.25 μm) was used to separate the reaction mixtures. The temperature of the column was initially increased from 80 to 220 °C at a rate of 20 °C min$^{-1}$ and held for 7 min. Naphthalene was used as the internal standard.

**Data availability.** The data that support the findings of this study are available from the corresponding author upon request.

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

## Acknowledgements
This work was supported by the National Natural Science Foundation of China (projects 21233008, 21703236, and 21790331), the Strategic Priority Research Program of the Chinese Academy of Sciences (No. XDB17020300), the Outstanding Young Scientist Foundation, the Chinese Academy of Sciences (CAS) and the Dalian Young Star of Science and Technology Project (No. 2016RQ027).

## Author contributions
X.J. designed the project and performed the catalyst preparation, characterizations, and tests and completed the paper. F.X, Y.X., and J.G. participated in beneficial discussions. J. M. and J.X. proposed, planned, designed, and supervised the project. All authors reviewed and commented on the manuscript.

## Additional information

**Competing interests:** The authors declare no competing interests.

