## [Peer Review File · Nature Communications]

Reviewers' comments:

Reviewer #1 (Remarks to the Author):

This manuscript is an interesting contribution to efforts aimed at controlling catalyst selectivity with organic modifiers. Here, the authors have used a novel type of modification, employing carboxylic acids to MnOx catalysts. They find that the carboxylic acids block sites associated with reaction of hydroxyl groups, allowing preservation of those groups during ammoxidation reactions. The authors demonstrate the basic mechanism for suppression of hydroxyl reaction via a nice set of infrared spectroscopy experiments. Overall, I think the manuscript is publishable in Nature Communications, though I do have a few suggestions for the manuscript:

- (1) Probably the most significant issue with the manuscript was that reaction kinetics weren't really reported in much detail, but instead the authors report conversions (as in Figure 5). While it is certainly important that the modified catalysts can achieve high selectivity at nearly full conversion, it would be useful to know how the modifiers affect the rate constants for the reactions. This is of practical importance, but also would provide additional insights on the catalysis. For example, does the rate of the desired reaction actually increase for the modified catalyst, or is it simply a case of selective poisoning?
- (2) I am not expert in manganese oxide surface chemistry, but I found it somewhat surprising that monodentate carboxylic acid adsorption was favored at low coverage, while the bidentate structure became populated at high coverage. Intuitively, I would expect the opposite trend, where the less crowded surface allows for more points of coordination with the acid. Is there any precedent for the trend proposed by the authors on related metal oxides? If not, are there other possible interpretations of these spectra?
- (3) Have the authors attempted to recycle their material, or conducted experiments in such a way that it can be clearly demonstrated that the catalyst performance remains steady over time? Figure 7 seems to be aimed at this, but it wasn't clear to me that it showed negligible loss of activity during operation. One might expect carboxylates to become detached under reaction conditions, or to undergo some form of degradation.
- (4) Do the authors have a suggestion for why the carboxylic acids selectively poison reactions of the hydroxyl, but do not poison the desired ammoxidations?
- (5) The data in general (as in Figure 5 and Table 1) do not include estimates of uncertainties; can these be provided?
- (6) Minor point: the authors rightly point out that there have been few studies using organic modifiers on metal oxides, but there is one very recent publication on modification of TiO₂ for

selectivity enhancement: <http://dx.doi.org/10.1021/acscatal.7b02789>

(7) The English is comprehensible, but there are numerous places where there are grammatical mistakes. In addition, in one case “metal” was misspelled as “meal”.

Reviewer #2 (Remarks to the Author):

The manuscript is dealing with the selective ammoxidation of 4-hydroxymethyl benzaldehyde and 5-hydroxymethyl furfural using MnOx modified with carboxylic acids. Here, it is very important to suppress the oxidation of OH group, and the strategy of the catalyst design is the poisoning of the oxidation site with the adsorption of carboxylic acids.

The present work may include interesting results, however, the demonstration is not appropriate. At first, authors should show the potential of MnOx without modification. For example, authors should show the results of MnOx at lower reaction temperature because the activity of MnOx is much higher than carboxylic acid-modified catalysts. At the same time, authors had better show the results of the activity test below 100% conversion. In particular, the activity of MnOx without and with the modification was so different, and the selectivity should not be discussed from the data around 100%. Authors should show the highest yield obtained on MnOx without modification by adjusting the reaction conditions such as reaction temperature, and reaction time, the ratio of the substrate to catalyst.

Another important point is the low activity of MnOx modified with carboxylic acids. Under the present reaction conditions, the turnover number per total Mn amount is calculated to be about only 3. Authors should demonstrate the data of much higher TON because it is very easy using much higher ratio of the substrate to catalyst, or reusability of the catalyst. In particular, in the present work, the catalyst stability and reusability should be demonstrated.

Moreover, authors should determine the amount of the adsorbed carboxylic acid during the reaction. Authors used the excess of carboxylic acids. How large is the amount of carboxylic acid in the solution? The amount of the adsorbed carboxylic acid may be influenced by the presence of NH₃ in the reaction system, and so on.

At present, it is very difficult to recommend for the publication.

Reviewer #3 (Remarks to the Author):

This paper is very poorly written. There are problems with grammar, words that make no sense, and other significant problems with the use of English. The paper needs to be rewritten by someone who knows the science of the work and English. In addition, there are scientific issues. Benzyl alcohol oxidation is easy to drive. The data are all > 99% conversion signifying equilibrium not catalysis. Figure 5 needs error bars on all data points. Figure 7 needs error bars

on all data points. If the product is chiral then either the surface or the reactant need to be chiral. This does not appear to be the case.

Responses to Reviewers:

Referee: 1

This manuscript is an interesting contribution to efforts aimed at controlling catalyst selectivity with organic modifiers. Here, the authors have used a novel type of modification, employing carboxylic acids to MnO_x catalysts. They find that the carboxylic acids block sites associated with reaction of hydroxyl groups, allowing preservation of those groups during ammoxidation reactions. The authors demonstrate the basic mechanism for suppression of hydroxyl reaction via a nice set of infrared spectroscopy experiments. Overall, I think the manuscript is publishable in Nature Communications, though I do have a few suggestions for the manuscript:

Comment 1: Probably the most significant issue with the manuscript was that reaction kinetics weren't really reported in much detail, but instead the authors report conversions (as in Figure 5). While it is certainly important that the modified catalysts can achieve high selectivity at nearly full conversion, it would be useful to know how the modifiers affect the rate constants for the reactions. This is of practical importance, but also would provide additional insights on the catalysis. For example, does the rate of the desired reaction actually increase for the modified catalyst, or is it simply a case of selective poisoning?

Response: Thank you for the reviewer's comments and nice suggestions. While it is significant to achieve high selectivity at nearly full conversion, the reaction kinetics should be investigated to provide insights on the ammoxidation activity of unmodified and modified catalyst. Aerobic oxidation reaction over manganese oxide catalyst is a complex cascade reaction without simple order; therefore we were not successful in predicting the rate constant. Instead, the time course of ammoxidation of 4-hydroxymethylbenzaldehyde on MnO_x and HAc modified MnO_x has been studied, respectively. As shown in Fig. R1a, the initial stage of reaction gave comparable amount of hydroxynitrile and dinitrile on unmodified MnO_x . By contrast, dinitrile was not detected on HAc modified MnO_x (Fig. R1b, Fig. R1d), indicating an extremely low rate of dinitrile formation. However, the rate of hydroxynitrile production was slightly decreased after modification (Fig. R1c). Thus, the enhanced chemoselectivity was not because of an increase in the rate of the desired reaction but rather a result of

inhibition of the undesired reaction for the modified catalyst.

Fig. R1 (Fig. 7 in the manuscript) Time course of aerobic ammoxidation of 4-hydroxymethylbenzaldehyde. a MnO_x as catalyst. **b** HAc modified MnO_x as catalyst. **c** Time course of 2 formation over MnO_x and HAc modified MnO_x, respectively. **d** Time course of 3 formation over MnO_x and HAc modified MnO_x, respectively. Reaction conditions: 0.5 mmol 4-hydroxymethyl-benzaldehyde, 0.17 mmol MnO_x, HAc/MnO_x = 120 mol% for HAc modified MnO_x, 5 mL CH₃CN, 0.3 MPa NH₃, 0.3 MPa O₂, 80 °C.

Comment 2: I am not expert in manganese oxide surface chemistry, but I found it somewhat surprising that monodentate carboxylic acid adsorption was favored at low coverage, while the bidentate structure became populated at high coverage. Intuitively, I would expect the opposite trend, where the less crowded surface allows for more points of coordination with the acid. Is there any precedent for the trend proposed by the authors on related metal oxides? If not, are there other possible interpretations of these spectra?

Response: It was previously reported that the adsorption of carboxylate on TiO₂ as bridging and monodentate binding modes are both energetically quite favorable, with small difference in adsorption energies: the bridging mode is typically about 10-25 kJ/mol more stable (Lars Ojamäe, et al., Journal of colloid and interface science, 2006, 296(1): 71-78). Nevertheless, carboxylate preferentially binds to the highly coordinative unsaturated sites on metal oxides through a single metal-O bond with

high strength, resulting in kinetically favored monodentate configuration; and due to the presence of the H-bond between the surface metal-OH and the O of the carboxylate binding to the metal, rotation of carboxylate or transformation to other mode is hindered (Gong, Xueqing, et al., Journal of Physical Chemistry B, 2006, 110(6): 2804-2811). Herein, highly coordinative unsaturated sites should be available at low coverage for kinetically favored monodentate mode adsorption on MnO_x .

Comment 3: Have the authors attempted to recycle their material, or conducted experiments in such a way that it can be clearly demonstrated that the catalyst performance remains steady over time? Figure 7 seems to be aimed at this, but it wasn't clear to me that it showed negligible loss of activity during operation. One might expect carboxylates to become detached under reaction conditions, or to undergo some form of degradation.

Response: We attempted to extend the reaction time to 18 h, and found that the selectivity for hydroxynitrile was well preserved with just slight decrease (Fig. R2). The capability of preventing alcohol ammoxidation over a long period suggests that the regulated chemisorption properties on the carboxylate modified catalyst surface cannot be easily destroyed under the reaction conditions. Moreover, the catalyst can be recycled without selectivity loss by regeneration (Fig. R3).

Fig. R2 (Fig. 8 in the manuscript) Time course of catalytic conversion of 4-hydroxymethylbenzaldehyde to 4-cyanobenzyl alcohol over HAc modified MnO_x . Reaction conditions: 2.5 mmol 4-hydroxymethyl-benzaldehyde, 0.85 mmol MnO_x , $\text{HAc}/\text{MnO}_x = 300$ mol%, 25 mL CH_3CN , 0.3 MPa NH_3 , 0.3 MPa O_2 , 80 °C.

Fig. R3 Recyclability test results for selective ammoxidation of 4-hydroxymethylbenzaldehyde catalyzed by HAc modified MnO_x . Reaction conditions: 0.5 mmol 4-hydroxymethyl-benzaldehyde, 0.17 mmol MnO_x , $\text{HAc}/\text{MnO}_x = 120$ mol%, 5 mL CH_3CN , 0.3 MPa NH_3 , 0.3 MPa O_2 , 80 °C, 3 h. At the completion of the reaction, the solution was separated by centrifugation and analyzed by GC using the internal standard method. The catalyst was washed with CH_3CN 3 times and then regenerated by adding HAc ($\text{HAc}/\text{MnO}_x = 120$ mol%) to the reaction mixture for the cycle experiments.

Comment 4: Do the authors have a suggestion for why the carboxylic acids selectively poison reactions of the hydroxyl, but do not poison the desired ammoxidations?

Fig. R4 (Fig. 2c in the manuscript) CH_3OH adsorption FT-IR spectra of unmodified and HAc-modified manganese oxides.

Response: To examine the effect of carboxylic acid modification on the

chemisorption behavior of alcohol on manganese oxide, in situ CH₃OH adsorption Fourier transform infrared spectroscopy (FT-IR) characterization (Fig. R4) was investigated. Methanol is dissociated into methoxy species on unmodified MnO_x and the bands of methoxy species at 1090 and 1028 cm⁻¹ assigned to the ν(C-O) were observed. Under equivalent conditions, the C-O vibrations arising from methoxy species were barely observed over HAc-modified MnO_x. The above results indicated that carboxylic acid can occupy the adsorptive sites for alcohol and the high stability of carboxylate adsorption hindered the potential replacement by alcohol. The higher adsorption stability of carboxylic acid compared with alcohol on metal oxides was also confirmed by density functional theory study (Gong, Xueqing, et al., Journal of Physical Chemistry B, 2006, 110(6): 2804-2811). In this way, the carboxylic acid modification poisoned the reaction of hydroxyl group. The aldehyde ammoxidation was not obviously affected, possibly because of the much different activation manner of C=O in aldehyde and C=NH in aldimine intermediate on metal oxides in comparison with alcohol.

Comment 5: The data in general (as in Figure 5 and Table 1) do not include estimates of uncertainties; can these be provided?

Fig. R5 (Fig. 5 in the manuscript) Performance of manganese oxides with different modifiers in ammoxidation of 4-hydroxymethylbenzaldehyde. Reaction conditions: 0.5 mmol 4-hydroxymethylbenzaldehyde, 0.17 mmol MnO_x, additive/MnO_x = 120 mol%, 5 mL CH₃CN, 0.3 MPa NH₃, 0.3 MPa O₂. 80 °C, 3 h.

Table R1 (Table 1 in the manuscript). Performance of manganese oxides with different amount of HAc modifiers in ammoxidation of 4-hydroxymethylbenzaldehyde.^a				
Entry	HAc/MnO_x (mol%)	Conv. (%)	Select. (%)	
			2	3
1 ^b	0	> 99	0	81 ± 1
2	6	> 99	12 ± 3	66 ± 1
3	12	> 99	45 ± 1	41 ± 3
4	30	> 99	71 ± 2	25 ± 2
5	120	> 99	91 ± 1	2 ± 1
6	300	> 99	92 ± 1	1 ± 0.4

^aReaction conditions: 0.5 mmol 4-hydroxymethyl-benzaldehyde, 0.17 mmol MnO_x, 5 mL CH₃CN, 0.3 MPa NH₃, 0.3 MPa O₂. 80 °C, 3 h. ^b18 ± 1% 4-Cyanobenzamide was detected.

Response: As shown in Fig. R5, table R1 and Fig. R2 and the revised manuscript, the estimates of uncertainties have been added to the data in Fig. 5, Table 1 and Fig. 8.

Comment 6: Minor point: the authors rightly point out that there have been few studies using organic modifiers on metal oxides, but there is one very recent publication on modification of TiO₂ for selectivity enhancement: <http://dx.doi.org/10.1021/acscatal.7b02789>

Response: This excellent work on utilization of organic modifiers for controlling the surface reactivity of titania has been added to the reference list.

Comment 7: The English is comprehensible, but there are numerous places where there are grammatical mistakes. In addition, in one case “metal” was misspelled as “meal”.

Response: We are sorry for the mistakes. We corrected the misspelled word. In addition, we improved our written English via “English Language Editing service” provided by Springer Nature.

Referee: 2

The manuscript is dealing with the selective ammoxidation of 4-hydroxymethyl benzaldehyde and 5-hydroxymethyl furfural using MnO_x modified with carboxylic acids. Here, it is very important to suppress the oxidation of OH group, and the strategy of the catalyst design is the poisoning of the oxidation site with the adsorption of carboxylic acids.

Comment 1: The present work may include interesting results, however, the demonstration is not appropriate. At first, authors should show the potential of MnO_x without modification. For example, authors should show the results of MnO_x at lower reaction temperature because the activity of MnO_x is much higher than carboxylic acid-modified catalysts. At the same time, authors had better show the results of the activity test below 100% conversion. In particular, the activity of MnO_x without and with the modification was so different, and the selectivity should not be discussed from the data around 100%. Authors should show the highest yield obtained on MnO_x without modification by adjusting the reaction conditions such as reaction temperature, and reaction time, the ratio of the substrate to catalyst.

Response: Thank you for the reviewer's comments and nice suggestions. The catalytic ammoxidation reaction over MnO_x has been studied by adjusting the reaction conditions like reaction time (Fig. R6), reaction temperature (Fig. R7) and ratio of the substrate to catalyst (Fig. R8). However, in spite of the reduced reaction time, decreased reaction temperature or decreased amount of catalyst, the chemoselective ammoxidation product was observed at low selectivity. Approximately 30% of the chemoselective hydroxynitrile ammoxidation product was obtained as the highest yield.

Besides the results at 100% conversion, we further studied the catalytic selectivity at 90%, 70% and 50% conversions, respectively, to obtain a better understanding of the role of the carboxylic acid in the ammoxidation reaction catalyzed by MnO_x (Fig. R9). And a steadily increased selectivity for hydroxynitrile was observed with increasing the carboxylate concentration at 90%, 70% and 50% conversions, respectively. This is in accordance with the change of selectivity at 100%

conversion (Table R1). By the way, a relatively low selectivity at 50% and 70% conversions was observed, which should be attributed to the formation of aldimine intermediate.

The reaction kinetics was also investigated to provide insights on the ammoxidation activity of catalyst. As shown in Fig. R1a, the initial stage of reaction gave comparable amount of hydroxynitrile and dinitrile on unmodified MnO_x . By contrast, dinitrile was not detected on HAc modified MnO_x (Fig. R1b, Fig. R1d), indicating an extremely low rate of dinitrile formation. However, the rate of hydroxynitrile production was slightly decreased after modification (Fig. R1c). Thus, the enhanced chemoselectivity was not because of an increase in the rate of the desired reaction but rather a result of inhibition of the undesired reaction for the modified catalyst.

Fig. R6 Time course of aerobic ammoxidation of 4-hydroxymethylbenzaldehyde over MnO_x at 80 °C. Reaction conditions: 0.5 mmol 4-hydroxymethyl-benzaldehyde, MnO_x /substrate = 34 mol%, 5 mL CH_3CN , 0.3 MPa NH_3 , 0.3 MPa O_2 , 80 °C.

Fig. R7 Time course of aerobic ammoxidation of 4-hydroxymethylbenzaldehyde over MnO_x at 80 °C.

MnO_x at 50 °C. Reaction conditions: 2.5 mmol 4-hydroxymethyl-benzaldehyde, MnO_x/substrate = 34 mol%, 25 mL CH₃CN, 0.3 MPa NH₃, 0.3 MPa O₂, 50 °C.

Fig. R8 Time course of aerobic ammoxidation of 4-hydroxymethylbenzaldehyde over MnO_x. Reaction conditions: 2.5 mmol 4-hydroxymethyl-benzaldehyde, MnO_x/substrate = 17 mol%, 25 mL CH₃CN, 0.3 MPa NH₃, 0.3 MPa O₂, 50 °C.

Fig. R9 Selectivity toward 4-cyanobenzyl alcohol for the ammoxidation of 4-hydroxymethylbenzaldehyde on MnO_x as a function of HAc/MnO_x. Reaction conditions: 2.5 mmol 4-hydroxymethyl-benzaldehyde, MnO_x/substrate = 34 mol%, 5 mL CH₃CN, 0.3 MPa NH₃, 0.3 MPa O₂, 80 °C.

Comment 2: Another important point is the low activity of MnO_x modified with carboxylic acids. Under the present reaction conditions, the turnover number per total Mn amount is calculated to be about only 3. Authors should demonstrate the data of much higher TON because it is very easy using much higher ratio of the substrate to catalyst, or reusability of the catalyst. In particular, in the present work, the catalyst stability and reusability should be demonstrated.

Response: We attempted to recycle the catalyst. As shown in Fig.R3, the catalyst can

be recycled for at least 9 times without selectivity loss by regeneration.

Fig. R3 Recyclability test results for selective ammoxidation of 4-hydroxymethylbenzaldehyde catalyzed by HAc modified MnO_x . Reaction conditions: 0.5 mmol 4-hydroxymethyl-benzaldehyde, $\text{MnO}_x/\text{substrate} = 34$ mol%, $\text{HAc}/\text{MnO}_x = 120$ mol%, 5 mL CH_3CN , 0.3 MPa NH_3 , 0.3 MPa O_2 , 80 °C, 3 h. At the completion of the reaction, the solution was separated by centrifugation and analyzed by GC using the internal standard method. The catalyst was washed with CH_3CN 3 times and then regenerated by adding HAc ($\text{HAc}/\text{MnO}_x = 120$ mol%) to the reaction mixture for the cycle experiments.

Comment 3: Moreover, authors should determine the amount of the adsorbed carboxylic acid during the reaction. Authors used the excess of carboxylic acids. How large is the amount of carboxylic acid in the solution? The amount of the adsorbed carboxylic acid may be influenced by the presence of NH_3 in the reaction system, and so on.

Response: Thank you for the reviewer's comments. It is difficult to measure the amount of the real-time adsorbed carboxylic acid during the reaction. Though, we measured the adsorbing capacity of MnO_x at 30 °C using n-hexanoic acid as probing molecule considering its similar effect to HAc (Fig. R5) and convenience in analysis by GC. The adsorptive property of carboxylic acid on MnO_x was investigated at various concentrations (Fig. R10). The result showed that the adsorption of carboxylic acid on MnO_x was nearly complete in the range of 0-12 mol% of n-hexanoic acid and increased slowly in range above 12 mol% which indicated the excess of unadsorbed carboxylic acids in the solution (Fig. R10).

It is reasonable to expect the reaction of carboxylic acid in the solution with NH_3 to form ammonium acetate. Ammonium acetate, which is strong electrolyte, should also have the capacity to release carboxylate to modify the catalyst surface. As shown in Table R2, ammonium acetate showed comparable effect on the MnO_x catalyzed selective ammoxidation of 4-hydroxymethylbenzaldehyde in comparison with HAc. Thus we can conclude that the reaction of NH_3 in the reaction system with carboxylic acid has little effect on the carboxylate adsorption and modification on MnO_x .

Fig. R10 Fig. 4. The chart of the n-hexanoic acid adsorption on MnO_x (30 °C).

Table R2. Performance of manganese oxides with ammonium acetate modifier in ammoxidation of 4-hydroxymethylbenzaldehyde. ^a				
Entry	Modifier	Conv. (%)	Select. (%)	
			2	3
1	Ammonium acetate	> 99	86	2
^a Reaction conditions: 0.5 mmol 4-hydroxymethyl-benzaldehyde, 0.17 mmol MnO_x , ammonium acetate/ MnO_x = 120 mol%, 5 mL CH_3CN , 0.3 MPa NH_3 , 0.3 MPa O_2 , 80 °C, 3 h.				

Referee: 3

Comment : This paper is very poorly written. There are problems with grammar, words that make no sense, and other significant problems with the use of English. The paper needs to be rewritten by someone who knows the science of the work and English. In addition, there are scientific issues. Benzyl alcohol oxidation is easy to drive. The data are all > 99% conversion signifying equilibrium not catalysis. Figure 5 needs error bars on all data points. Figure 7 needs error bars on all data points. If the product is chiral then either the surface or the reactant need to be chiral. This does not appear to be the case.

Response: Thanks for the reviewer's comments and suggestions. We improved our written English via "English Language Editing service" provided by Springer Nature. The selectivity under lower conversion has been studied (Fig. R9). A steadily increased selectivity for hydroxynitrile was observed with increasing the carboxylate concentration at 90%, 70% and 50% conversions. This is in accordance with the change of selectivity at 100% conversion (Table R1).

The reaction kinetics was also investigated to provide insights on the ammoxidation activity of catalyst. As shown in Fig. R1a, the initial stage of reaction gave comparable amount of hydroxynitrile and dinitrile on unmodified MnO_x . By contrast, dinitrile was not detected on HAc modified MnO_x (Fig. R1b, Fig. R1d), indicating an extremely low rate of dinitrile formation. However, the rate of hydroxynitrile production was slightly decreased after modification (Fig. R1c). Thus, the enhanced chemoselectivity was not because of an increase in the rate of the desired reaction but rather a result of inhibition of the undesired reaction for the modified catalyst.

In addition, as shown in Fig. R2 and Fig. R5, we added error bars after repeat measurements. Organic modification with chiral modifiers is a powerful tool to enhance the chiral selectivity. While, herein the ammoxidation product is not chiral. Controlling chiral selectivity in ammoxidation reaction is potentially significant work, which deserves study in the future.

Fig. R9 Selectivity toward 4-cyanobenzyl alcohol for the ammoxidation of 4-hydroxymethylbenzaldehyde on MnO_x as a function of HAc/MnO_x. Reaction conditions: 2.5 mmol 4-hydroxymethyl-benzaldehyde, MnO_x/substrate = 34 mol%, 5 mL CH₃CN, 0.3 MPa NH₃, 0.3 MPa O₂, 80 °C.

Fig. R1 (Fig. 7 in the manuscript) Time course of aerobic ammoxidation of 4-hydroxymethylbenzaldehyde. **a** MnO_x as catalyst. **b** HAc modified MnO_x as catalyst. **c** Time course of 2 formation over MnO_x and HAc modified MnO_x, respectively. **d** Time course of 3 formation over MnO_x and HAc modified MnO_x, respectively. Reaction conditions: 0.5 mmol 4-hydroxymethyl-benzaldehyde, 0.17 mmol MnO_x, HAc/MnO_x = 120 mol% for HAc modified MnO_x, 5 mL CH₃CN, 0.3 MPa NH₃, 0.3 MPa O₂, 80 °C.

Fig. R2 (Fig. 8 in the manuscript) Time course of catalytic conversion of 4-hydroxymethylbenzaldehyde to 4-cyanobenzyl alcohol over HAc modified MnO_x . Reaction conditions: 2.5 mmol 4-hydroxymethyl-benzaldehyde, 0.85 mmol MnO_x , $\text{HAc}/\text{MnO}_x = 300$ mol%, 25 mL CH_3CN , 0.3 MPa NH_3 , 0.3 MPa O_2 , 80 °C.

Fig. R5 (Fig. 5 in the manuscript) Performance of manganese oxides with different modifiers in ammoxidation of 4-hydroxymethylbenzaldehyde. Reaction conditions: 0.5 mmol 4-hydroxymethylbenzaldehyde, 0.17 mmol MnO_x , additive/ $\text{MnO}_x = 120$ mol%, 5 mL CH_3CN , 0.3 MPa NH_3 , 0.3 MPa O_2 , 80 °C, 3 h.

Reviewers' comments:

Reviewer #1 (Remarks to the Author):

I think the authors have done a good job of responding to the reviews, and their paper is now publishable in Nature Communications.

Reviewer #2 (Remarks to the Author):

The manuscript was revised appropriately.

Reviewer #3 (Remarks to the Author):

Some of the suggestions have been followed including a rewrite of the paper. There are new kinetic data as well.